# Growth inhibition associated with disruption of the actin cytoskeleton by Latrunculin A in rhabdomyosarcoma cells

Julia Würtemberger[1], Daria Tchessalova[2], Carla Regina[1], Christoph Bauer[1], Michaela Schneider[1], Amy J. Wagers[2,3], Simone Hettmer[1] *

1 Division of Pediatric Hematology and Oncology, Department of Pediatric and Adolescent Medicine, University Medical Center Freiburg, University of Freiburg, Freiburg, Germany, 2 Joslin Diabetes Center, Boston, Massachusetts, United States of America, 3 Department of Stem Cell and Regenerative Biology, Harvard University and Harvard Stem Cell Institute, Cambridge, Massachusetts, United States of America

* simone.hettmer@uniklinik-freiburg.de

**Data Availability Statement:** All relevant data are within the manuscript and its supporting information files.

## Abstract

Functional genomic screening of KRAS-driven mouse sarcomas was previously employed to identify proliferation-relevant genes. Genes identified included Ubiquitin-conjugating enzyme E2 (*Ube2c*), Centromere Protein E (*Cenpe*), Hyaluronan Synthase 2 (*Has2*), and CAMP Responsive Element Binding Protein 3 Like 2 (*Creb3l2*). This study examines the expression and chemical inhibition of these candidate genes, identifying variable levels of protein expression and significant contributions to rhabdomyosarcoma (RMS) cell proliferation. Chemical treatment of human and murine RMS cell lines with bortezomib, UA62784, latrunculin A and sorafenib inhibited growth with approximate EC50 concentrations of 15-30nM for bortezomib, 25-80nM for UA62784 and 80-220nM for latrunculin A. The multi-kinase inhibitor sorafenib increased *in vitro* proliferation of 4 of 6 sarcoma cell lines tested. Latrunculin A was further associated with disruption of the actin cytoskeleton and reduced ERK1/2 phosphorylation. Together, this work advances opportunities for developing therapies to block progression of soft-tissue sarcomas and demonstrates that disruption of the actin cytoskeleton in sarcoma cells by latrunculin A is associated with a reduction in RMS cell growth. (*167 words*).

## Introduction

Rhabdomyosarcoma (RMS) represents the most common soft-tissue sarcoma (STS) subtype within the pediatric age group [1, 2]. RAS pathway genes are frequently mutated in STS [3], including PAX3/7:FOXO1 fusion-negative RMS [4]. We previously reported on a customized shRNA-based proliferation screen [5], which tested the contributions of 141 sarcoma-relevant genes, previously identified by transcriptional profiling of genetically engineered mouse sarcomas driven by *KRAS(G12v)* and CDKN2A deletion [6]. In this screen, the strongest inhibitory effect on sarcoma growth was produced by silencing of asparagine synthetase (ASNS), which established that adequate asparagine availability was a metabolic vulnerability with potential

**Funding:** This work was funded by a Stand Up To Cancer-American Association for Cancer Research Innovative Research Grant (SU2CAACR- IRG1111; to AJW) and P.A.L.S. Bermuda/St. Baldrick's (to SH). We did not receive any other external funding to support this study.

**Competing interests:** The authors have declared that no competing interests exist.

anti-sarcoma therapeutic value [5]. The screen also identified four other potentially druggable genes/ cellular processes: (1) Ubiquitin-conjugating enzyme E2 (*Ube2c*), which is essential in cell cycle progression by orchestrating proteolysis of cyclin-dependent kinase and its inhibitors [7]; (2) Centromere Protein E (*Cenpe*), a kinesin-like protein that localizes to the kinetochore during mitosis and is important for bipolar spindle formation [8]; (3) Hyaluronan Synthase 2 (*Has2*), responsible for the synthesis of hyaluronan, which serves as a scaffold for the extracellular matrix and critically determines the extracellular micromilieu [9]; and, (4) CAMP Responsive Element Binding Protein 3 Like 2 (*Creb3l2*), a transcription factor and downstream target of mitogen-activated (MAPK) signalling that promotes tumor cell survival [10]. This study aimed at determining the expression of UBE2C, CENPE, HAS2 and CREB3L2 in human STS and examining the growth-attenuating effects of candidate chemicals bortezomib (aimed at UBE2C; [7]), UA62784 (aimed at CENPE; [8]), sorafenib (aimed at Creb3l2; [10]) and latrunculin A (aimed at HAS2; [9]) on the *in vitro* growth of human and murine sarcomas. Findings from our experiments highlight that inhibition of actin polymerization by latrunculin A is linked to reduced growth of RMS cells.

## Materials and methods

### Sarcoma cell lines

Mouse sarcoma cell lines were derived from a *Kras;CDKN2A$^{null}$* mouse sarcoma with myogenic differentiation (RMS) and a *Kras;CDKN2A$^{null}$* undifferentiated, non-myogenic mouse sarcoma (NMS). The human RMS cell line RD (PAX3/7:FOXO1-negative) and the human fibrosarcoma line HT1080 originated from ATCC. Human RMS cell lines Rh3, Rh5, Rh10, Rh28, Rh30, Rh41 (all PAX3:FOXO1-positive) and Rh36 (PAX3/7:FOXO1-negative) were gifts from Dr. Peter Houghton (Greehey Children's Research Institute, San Antonio, TX, USA). All cell lines were grown in DMEM with 10% FBS and 1% Penicillin-Streptomycin.

### Customized shRNA proliferation screen

The shRNA proliferation screen was carried out in two *Kras;CDKN2A$^{null}$* mouse sarcoma cell lines as previously described [5]. The screen and details of the statistical analysis were published previously [5]. In brief, each candidate gene was targeted by 5 individual shRNAs. For each shRNA, relative cell proliferation was determined as the percentage growth of shRNA infected cells compared to the mean growth of cells infected with cntrl-shRNAs. Differences in average proliferation between cells infected with shRNAs against one specific target gene and average proliferation of cntrl-shRNA infected cells were tested for statistical significance using T-tests and the algorithm published by J. W. McNicol and G. Hogan [11]. Receiver operator curve analysis established a false discovery rate less than 30% for relative proliferation of less than 52% or 40% of cntrl-shRNA infected cells for the two lines. The growth-inhibitory effects of shRNA-mediated silencing of individual candidate genes were considered significant if $p < 0.01$ and $q < 0.05$ and 3 shRNAs scored with an FDR < 30%.

### Immunohistochemistry

Candidate gene expression in primary human sarcoma tissue was evaluated using commercially available sarcoma tissue arrays (US Biomax SO2081). Paraffin was removed by placing slides in a Coplin jar at 58 degrees centigrade in a microwave oven. Slides were then rehydrated by immersing them serially in xylene (3 x 5 minutes), 90% ethanol (1 x 3 minutes) and 80% ethanol (1 x 3 minutes) prior to rinsing them in gently running tap water and placing them in PBS for 30 minutes. Antigen retrieval was performed in 10mM sodium citrate buffer

pH6 in a microwave oven operated at high power for 5 minutes, and tissue sections were blocked in PBS, 5% BSA, pH7.4. Tissue was stained for CENPE (1 in 500, HPA042294, Sigma; human testis served as positive and brain as negative control tissue), UBE2C (1 in 200, A-650, Boston Biochem Inc; colon served as positive and brain as negative control tissue), CREB3L2 (1 in 200, HPA015068, Sigma; liver served as positive and colon as negative control tissue) and HAS2 (1 in 600, ab140671, Abcam; dermis served as positive and brain as negative control tissue). Primary antibodies were incubated overnight at 4 degree centigrade. Control tissues were obtained from the National Disease Research Interchange (NDRI) (S1 Fig). Primary antibody binding was detected by labeling with biotinylated secondary antibodies (1 in 800, B8895, Sigma) and Streptavidin-HRP (BD, 51-75477E). Slides were then exposed to DAB substrate (BD, 550880), which reacts with HRP to produce a brown-colored signal. CENPE (nuclear), UBE2C (cytoplasmic), CREB3L2 (cytoplasmic) and HAS2 (nuclear) staining was evaluated by two independent operators. If > 25% of cells per core exhibited a positive signal, antigen expression was considered positive.

## RNA isolation and qRT-PCR

RNA was isolated from human RD, HT1080, Rh3, Rh5, Rh10, Rh28, Rh30, Rh41 and Rh36 cells and murine SMPO1 cells by TRIzol extraction followed by DNAse digestion and purification using the RNeasy Plus Micro Kit. The use of human muscle as control tissue was approved by the Institutional Review Board at Joslin Diabetes Center. Human fetal muscle was obtained from 20–23 week gestation fetuses and adult muscle from deceased volunteers. Tissue was homogenized in TRIzol using a tissue homogenizer prior to RNA isolation as described above. RNA was reverse transcribed using Superscript III First-Strand Synthesis System for RT-PCR (Invitrogen). qRT-PCR was performed using an ABI 7900 RT-PCR system (Applied Biosystem) with SYBR-green PCR reagents.

*CENPE*, *HAS2*, *UBE2C* and *CREB3L2* in human tissue were detected using the following primer sequences: GATTCTGCCATACAAGGCTACAA (CENPE, fw); TGCCCTGGG-TATAACTCCCAA (CENPE, rev); CTCTTTTGGACTGTATGGTGCC (HAS2, fw), AGGG TAGGTTAGCCTTTTCACA (HAS2, rev); GACCTGAGGTATAAGC-TCTCGC (UBE2C, fw), TTACCCTGGG-TGTCCACGTT (UBE2C, rev); CAGAGAAGAGTGTGTCAATGGAG (CREB3L2, fw), CTGGTGGTAAT-GTGGGTGAAG (CREB3L2, rev).

*Cenpe*, *Has2*, *Ube2c* and *Creb3l2* in mouse tissue were detected using the following primer sequences: TCAGGAAAGACACACACGATG (Cenpe, fw); TGCGAGCCATTTCAAAGCCA (Cenpe, rev); TGTGAGAGGTTTCTATGTGTCCT (Has2, fw), ACCGTACAGTCCAAATGA GAAGT (Has2, rev); CTCCGCCTTCCCTGAGTCAGC (Ube2c, fw), GGTGCGTTGTAAGG GTAGCC (Ube2c, rev); CATGTACCACACGCACTTCTC (Creb3l2, fw), CCACCTCCATTG ACTCGCT (Creb3l2, rev).

## Proliferation assays

Mouse *Kras;CDKN2A*<sup>null</sup> SMP-01 RMS and Sca1-01 NMS, human RD, Rh30 and Rh41 RMS and human HT1080 fibrosarcoma cells were exposed to the following chemicals: sorafenib (0.1-1mM, stock 10mM in DMSO, Cayman Chemicals), bortezomib (50-250nM, stock 10mM in DMSO, Cayman Chemicals), latrunculin A (50-1000nM, stock 237 mM in ethanol, Cayman Chemicals), UA62784 (50-250nM, stock 5.66mM in DMSO, Sigma) and vehicle (DMSO, ethanol). Proliferation assays were performed as previously described [1, 6]. Estimated EC50 concentrations were calculated using GraphPad Prism.

## Western blot

Cells were washed with PBS and lysed in lysis buffer (New England Biolabs). Protein concentrations were determined using DC protein assays (Biorad). Membranes were incubated with primary antibodies against p44/42 MAPK (Erk1/2; titer 1:2000; 9102, Cell Signaling Technology), phospho-p44/42 MAPK (phospho-Erk1/2; titer 1:2000; 9101; Cell Signaling Technology) and GAPDH (titer 1:10000; 2118, Cell Signaling Technology) at 4 degrees overnight. Secondary antibodies (titer 1:10000; 170–6515, Biorad) were incubated for 1 hour at room temperature.

## Immunocytochemistry

Cells were fixed with 4% paraformaldehyde (PFA), incubated with triton X 0.2% and blocked with 10% goat serum 10%. Human cells were stained with Phalloidin iFluor 488 (titer 1:100, A12379, Thermo Scientific), and mouse cells were stained with Phalloidin iFluor 594 (titer 1:100, ab176757, Abcam) at room temperature for one hour. Nuclei were stained with 1μg/ml 4'-6-diamino-2-phenylindole (DAPI, D9542, Sigma Aldrich). Staining was then evaluated by immunofluorescence microscopy using a Zeiss LSM 710 confocal microscope. RD and Rh30 human RMS cells were cultured and stained in uncoated 96-well-dishes. *Kras;CDKN2A^null* mouse sarcoma cells were cultured and stained on 10% matrigel (354234, Corning).

## Statistics

Differences in cell growth were tested for statistical significance using T-tests (ns $p \geq 0.05$, * $p < 0.05$, ** $p < 0.01$, *** $p < 0.001$).

## Results

### Identification of target genes

A candidate set of 141 sarcoma-relevant genes (S1 Table), identified by transcriptional profiling of genetically engineered mouse RMS and non-myogenic sarcomas driven by *KRAS(G12v)* and CDKN2A deletion, was previously published [6]. All 141 candidate genes were tested by shRNS proliferation screening [6], designed to determine the growth-promoting effects of each of these candidates. The screen employed five shRNAs per target gene and two low-passage *Kras;CDKN2A^null* mouse sarcoma cell lines [5]. Sixteen of 141 candidate genes met significance criteria in one or both cell lines (Table 1; $p < 0.01$, $q < 0.05$, at least 3 shRNAs with FDR<30%; [5]). Published literature was reviewed to identify the genes, within this list of 16, for which chemical modulators had previously been described, because such small molecules could potentially be re-purposed as anti-sarcoma drugs. This review identified five potentially druggable genes/ cellular processes: ASNS, UBE2C, CENPE, HAS2, CREB3L2 (Fig 1A). As prior work already has validated asparagine starvation as an actionable metabolic vulnerability in sarcomas [5, 12], this study concentrates on the potential roles of UBE2C, CENPE, HAS2 and CREB3L2 in STS with a focus on RMS. Effective knockdown, confirmed by PCR (Fig 1B), of each of these targets by the target-specific shRNAs employed in our screen reduced cell proliferation to 40–60% of that seen in control cell cultures (Table 1).

### Candidate gene expression in sarcomas

Candidate proteins CENPE, CREB3L2, HAS2 and UBE2C were detected in primary human STS tissue at variable levels (Fig 2A and 2B). In human RMS, UBE2C was detected in 8 of 21 (38%), CENPE and HAS2 in 4 of 24 (17%) and CREB3L2 in 20 of 22 (91%) (Fig 2B). In human leiomyosarcomas (LMS), expression of UBE2C was found in 12 of 26 (46%), of CENPE in 7 of

**Table 1. Proliferation-relevant sarcoma genes.** Transcriptional profiling of genetically engineered, KRAS-induced mouse rhabdomyosarcomas (RMS) and non-myogenic sarcomas (NMS) identified 141 sarcoma-relevant genes; their function was evaluated using a customized shRBA screen. Sixteen of the 141 sarcoma-relevant genes scored as „hits"in RMS and/ or NMS ($p < 0,01$, $q > 0,05$, at least 3 of 5 shRNAs with FDR< 30%) and are listed here. These 16 hits include 5 potentially druggable targets, i.e. Asns, Cenpe, Crebl2, HAs2 and Ube2c.

| | | | Kras; $p16p19^{null}$ RMS | | | | Kras; $p16p19^{null}$ NMS | | | |
|---|---|---|---|---|---|---|---|---|---|---|
| Symbol | Cell function | Chemical modulation | Mean % of ctrl | Level | p-value | q value | Mean % of ctrl | Level | p-value | q value |
| **Asns** | Asparagine synthesis | AS5, Asparaginase, Mupirocin | 30,16 | 2 | < .0001 | 0,0058 | 6,6934 | 1 | < .0001 | 0,0072 |
| Rbbp8 | Cell cycle | | 56,06 | 3 | 0,0022 | 0,0160 | 30,7323 | 2 | 0,0012 | 0,0288 |
| Rad54l | DNA repair | | 57,56 | 3 | 0,0031 | 0,0180 | 40,5093 | 3 | 0,0076 | 0,0421 |
| **Ube2c** | Cell cycle | Bortezomib | 43,56 | 3 | < .0001 | 0,0058 | 38,6873 | - | 0,0054 | 0,0389 |
| **Cenpe** | Chromosome stability | UA62784, GSK923295A | 72,49 | - | 0,0511 | 0,0689 | 39,9054 | 3 | 0,0067 | 0,0402 |
| **Has2** | Extracellular matrix | Latrunculin A | 61,76 | - | 0,0075 | 0,0335 | 41,5132 | 3 | 0,0088 | 0,0453 |
| **Creb3l2** | Transcriptional regulation | Sorafenib | 62,69 | 2 | 0,0090 | 0,0326 | 60,2497 | - | 0,1230 | 0,1736 |
| Basp1 | Transcriptional regulation | | 45,37 | 3 | 0,0002 | 0,0039 | 50,8361 | - | 0,0371 | 0,0989 |
| Lrrfip1 | Transcriptional regulation | | 54,80 | 3 | 0,0017 | 0,0164 | 54,4497 | - | 0,0606 | 0,1322 |
| Lasp1 | Cytoskeletal organization | | 60,37 | 3 | 0,0056 | 0,0271 | 58,6252 | 3 | 0,1019 | 0,1706 |
| Pold3 | DNA repair | | 62,96 | 3 | 0,0095 | 0,0290 | 31,9654 | - | 0,0015 | 0,0216 |
| Efhd2 | Apoptosis | | 53,27 | - | 0,0034 | 0,0179 | 46,8886 | 2 | 0,0007 | 0,0252 |
| Myo9b | Cytoskeletal organization | | 67,41 | - | 0,0445 | 0,0737 | 33,5929 | 2 | 0,0021 | 0,0252 |
| Runx1 | Transcriptional regulation | | 64,64 | - | 0,0455 | 0,0713 | 30,6930 | 3 | 0,0012 | 0,0288 |
| Shcbp1 | Cell proliferation | | 71,42 | - | 0,1118 | 0,1099 | 37,7316 | 3 | 0,0045 | 0,0405 |
| Egr2 | Transcriptional regulation | | 74,44 | - | 0,0689 | 0,0850 | 42,9416 | 3 | 0,0048 | 0,0384 |

26 (27%), of HAS2 in 7 of 26 (25%) and of CREB3L2 in 20 of 27 (74%) human leiomyosarcoma cores (Fig 2B). CREB3L2 was also detected in normal human skeletal muscle samples (7 of 8 (88%) samples), but UBE2C, CENPE and HAS2 were not.

Expression levels of candidate genes were further evaluated in nine human sarcoma cell lines by RT-PCR. Increased expression of *UBE2C*, *CENPE* and *CREB3L2* compared to normal muscle was detected in 9 of 9 human sarcoma cell lines analyzed (S2A–S2C Fig), whereas *HAS2* expression was detected in only 5 of 9 human sarcoma lines (S2D Fig). *UBE2C*, *CENPE* and *CREB3L2* were also detected in normal human skeletal muscle, while *HAS2* was detected in fetal muscle only (S2D Fig). Discrepancies between candidate expression levels in sarcoma cell lines and primary tissue could be due to passaging *in vitro*.

## Anti-proliferative effects of chemical targeting of candidate genes

Given that downregulation of *Ube2c*, *Cenpe*, *Has2* and *Creb3l2* by shRNA knockdown diminished mouse sarcoma cell proliferation *in vitro* (Table 1), we hypothesized that small molecule inhibitors previously reported to impede the cellular functions of these targets could prove useful in inhibiting their growth. We obtained and tested 4 chemical compounds: (1) the proteasome inhibitor bortezomib, (2) the ATPase inhibitor UA62784, (3) the sponge-derived macrolide latrunculin A, and (4) the protein kinase inhibitor sorafenib. Bortezomib has been shown to downregulate UBE2C and mediate accumulation of cyclins A and B1 [7]. UA62784 and GSK923295A were shown to inhibit ATPase activity in the CENPE motor protein [8], although UA62784 has also been implicated with microtubule polymerization and associated with accumulation of mammalian cells in mitosis due to aberrant formation of mitotic spindles [13]. Latrunculin A was previously shown to reduce HAS2 expression in fibroblasts [9], and sorafenib was shown to reduce CREB3L2 expression in glioma cells [10]. Of the four chemicals tested, three inhibited proliferation of mouse *Kras;p16p19^{null}* RMS and NMS cells, human HT1080 fibrosarcoma cells, human fusion-negative RMS cells (RD) and two different

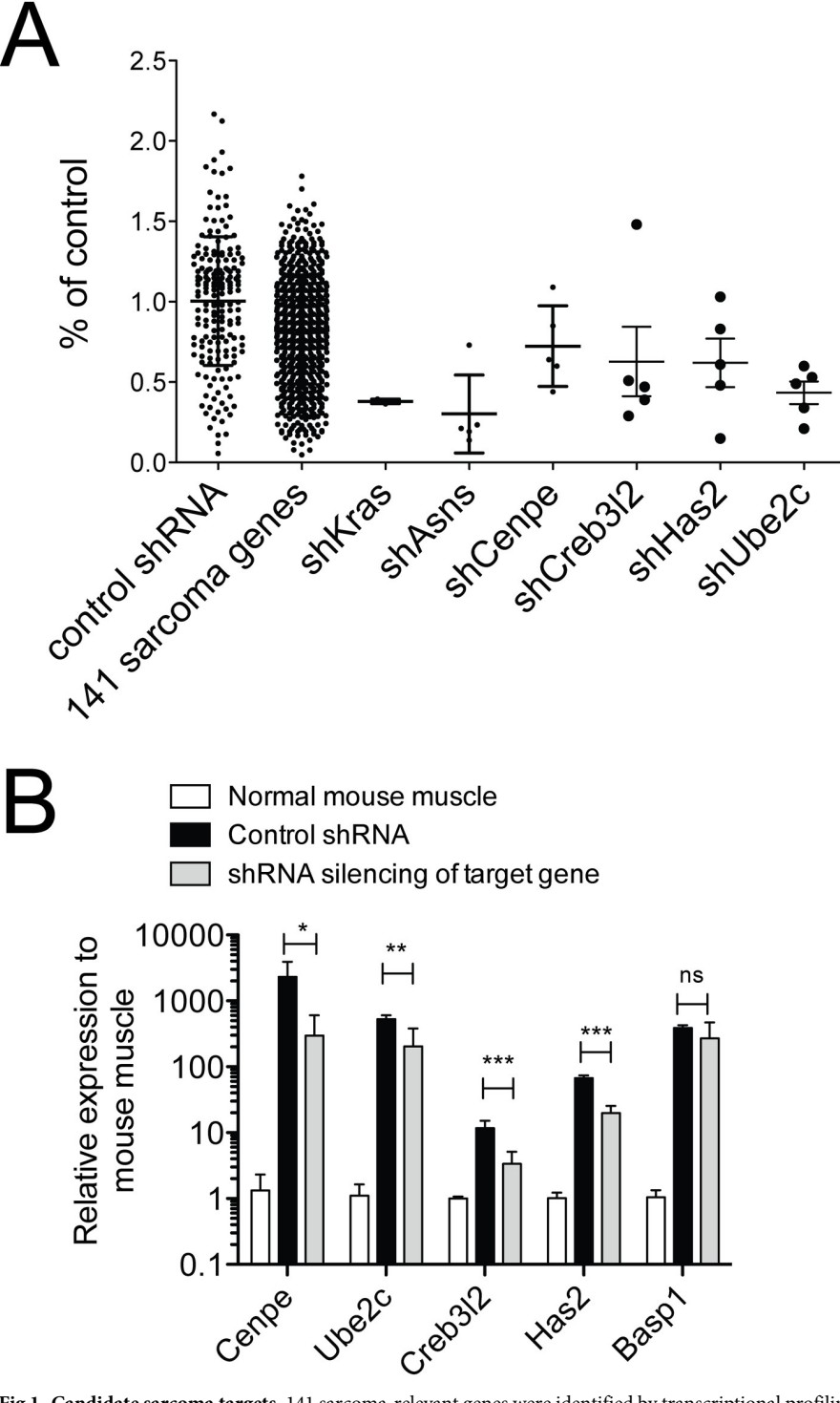

**Fig 1. Candidate sarcoma targets.** 141 sarcoma-relevant genes were identified by transcriptional profiling of genetically engineered *Kras*-driven mouse sarcomas [6]. Their contributions to sarcoma growth were probed by customized shRNA screening using 5 shRNAs per candidate gene. Control shRNAs were directed against LUC, RFP and LACZ; shKras served as a positive control [5]. (A) Five candidate genes whose targeting by shRNAs in this screen resulted in reduced sarcoma cell proliferation represent potentially druggable genes/ cellular processes: *Asns*, *Ube2c*, *Cenpe*, *Has2*, *Creb3l2*. The anti-proliferative effects of the shRNAs directed against these candidate genes in a *Kras; CDKN2A^{null}* mouse RMS cell line are shown. (B) Effective knockdown by the target-specific shRNAs in *Kras; CDKN2A^{null}* mouse RMS cells was confirmed by qRT-PCR. Target gene expression was determined by gene-specific qRT-PCR (mean +/- SD of 3 technical replicates presented; ns $p \geq 0.05$, $^{*}$ $p < 0.05$, $^{**}$ $p < 0.01$, $^{***}$ $p < 0.001$, as determined by T-tests compared to cntrl-shRNA infected cells).

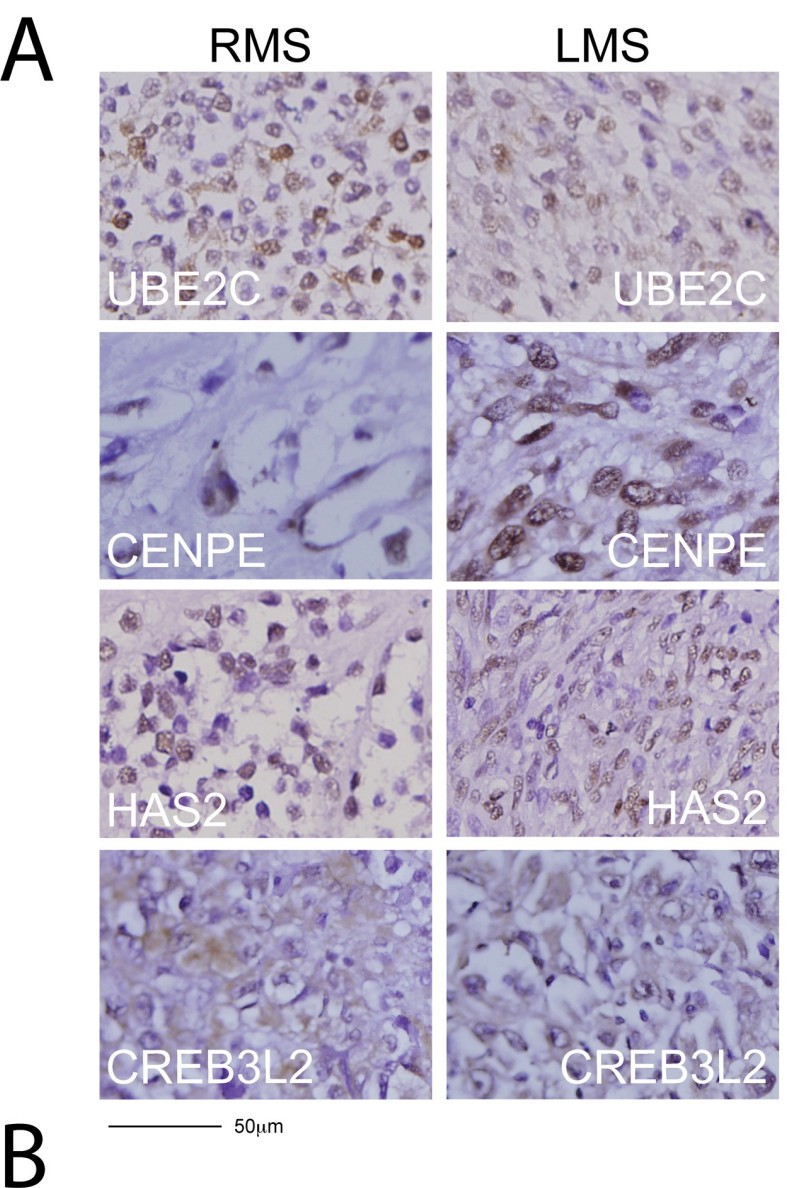

| Tissue | UBE2C positive | CENPE positive | HAS2 positive | CREB3l2 positive |
|---|---|---|---|---|
| Skeletal muscle | 0 of 8 (0%) | 0 of 8 (0%) | 0 of 7 (0%) | 7 of 8 (88%) |
| Leiomyosarcoma | 12 of 26 (46%) | 7 of 26 (27%) | 7 of 28 (25%) | 20 of 27 (74%) |
| Rhabdomyosarcoma | 8 of 21 (38%) | 4 of 24 (17%) | 4 of 24 (17%) | 20 of 22 (91%) |

**Fig 2. Expression of candidate sarcoma targets in human sarcoma tissue.** Immunohistochemical staining of commercially available sarcoma tissue arrays (US Biomax SO2081) was used to confirm candidate expression in human sarcomas. (A) Representative stains in RMS (left column) and high-grade LMS (right column) are shown for UBE2C (top row), CENPE (second row from top), HAS2 (third row from top) and CREB3L2 (bottom row). (B) Candidate proteins were expressed in 17–91% of RMS cores and 27–74% of LMS cores. Please see also S1 and S2 Figs.

PAX3:FOXO1-positive human RMS cell lines (Rh30, Rh41) (Fig 3A and 3B). Approximate EC50 concentrations were 15-30nM for bortezomib, 25-80nM for UA62784 and 80-220nM for latrunculin A (Fig 3B). Sorafenib had a slight inhibitory effect on NMS proliferation, with an estimated EC50 concentration of 200nM; yet, unexpectedly, sorafenib had no inhibitory effects on the other cell lines tested. In fact, sorafenib increased the growth of mouse RMS and of all but one of the human sarcoma cell lines tested (Fig 3A and 3B).

## Latrunculin A effects on RMS cells

Latrunculin A inhibited the growth of mouse and human RMS cell lines. Its effects on target gene expression were evaluated by qRT-PCR in mouse and human RMS cell lines. There was no clear effect on HAS2 expression in human RD, human Rh30 and mouse RMS cells (S3 Fig).

As latrunculin A is known to bind actin monomers and induce depolymerization of the cytoskeleton [14, 15], we evaluated the actin cytoskeleton in Latrunculin-A treated cells by phalloidin staining. Human RD sarcoma cells were exposed to 250nM latrunculin A, human Rh30 RMS cells to 100nM latrunculin A and mouse RMS sarcoma cells to 100nM latrunculin A. Different concentrations were chosen due to differences in latrunculin A sensitivity between sarcoma cell lines. There was profound disruption of the F-actin cytoskeleton after Latrunculin A treatment of all 3 RMS cell lines, including human RD (Fig 4B, to panels) and Rh30 cells (Fig 4B, middle panels) and mouse RMS cells (Fig 4B, bottom panels). Finally, we demonstrated diminished ERK1/2-phosphorylation in human and mouse RMS cells exposed to Latrunculin A (Fig 4A, S1 Raw images).

## Discussion

Radiation and surgery continue to be a mainstay in the treatment of STS, including RMS [16]. The outcome of tumors that have spread regionally and/or systemically is dismal [17]. Unlike many other STS tumors, RMS is relatively sensitive to conventional cytostatic drugs, and currently available multimodal treatment strategies cure approximately 65% of children and adolescents with RMS [18]. Still, more than 70% of those diagnosed with metastatic RMS die from disease [19]. These statistics highlight the need for new drugs to treat STS.

Functional genomic screening of KRAS-driven mouse sarcomas was employed to identify actionable proliferation-relevant genes of potential therapeutic applicability. Proliferation assays in 2 mouse and 4 human sarcoma cell lines, including one mouse and 3 human RMS cell lines, revealed growth inhibition by the proteasome inhibitor bortezomib, the CENPE inhibitor UA62784 and latrunculin A. Yet, interestingly, the multi-kinase inhibitor sorafenib increased *in vitro* proliferation of 5 of 6 sarcoma cell lines tested, including 3 human RMS cell lines. Bortezomib, CENPE inhibitors and sorafenib were previously evaluated by the Pediatric Preclinical Testing Program for anti-RMS effects as single agents. Bortezomib and sorafenib did not reduce the size of RMS xenografts (n = 6–7, [20, 21]). The CENPE inhibitor GSK923295A only induced an objective response in 2 of 5 xenografts tested [22]. Taken together, these *in vivo* observations do not support a major role for bortezomib, CENPE inhibitors and sorafenib in RMS treatment.

Latrunculin A is a 2-thiazolidinone macrolide derived from sponges that binds monomeric actin with 1:1 stoichiometry, blocks actin polymerization and results in depolymerization of tumor cell cytoskeleton. Antitumorigenic effects of Latrunculin A have been reported in prostate cancer, hepatocellular carcinoma and gastric cancer [23–25]. Latrunculin A was tested in this study, because it was previously shown to reduce *HAS2* expression in fibroblasts [9]. Yet, in our experiments, latrunculin A did not decrease *HAS2* expression in mouse or human RMS cells. However, we did observe a profound disruption of the actin cytoskeleton and reduced

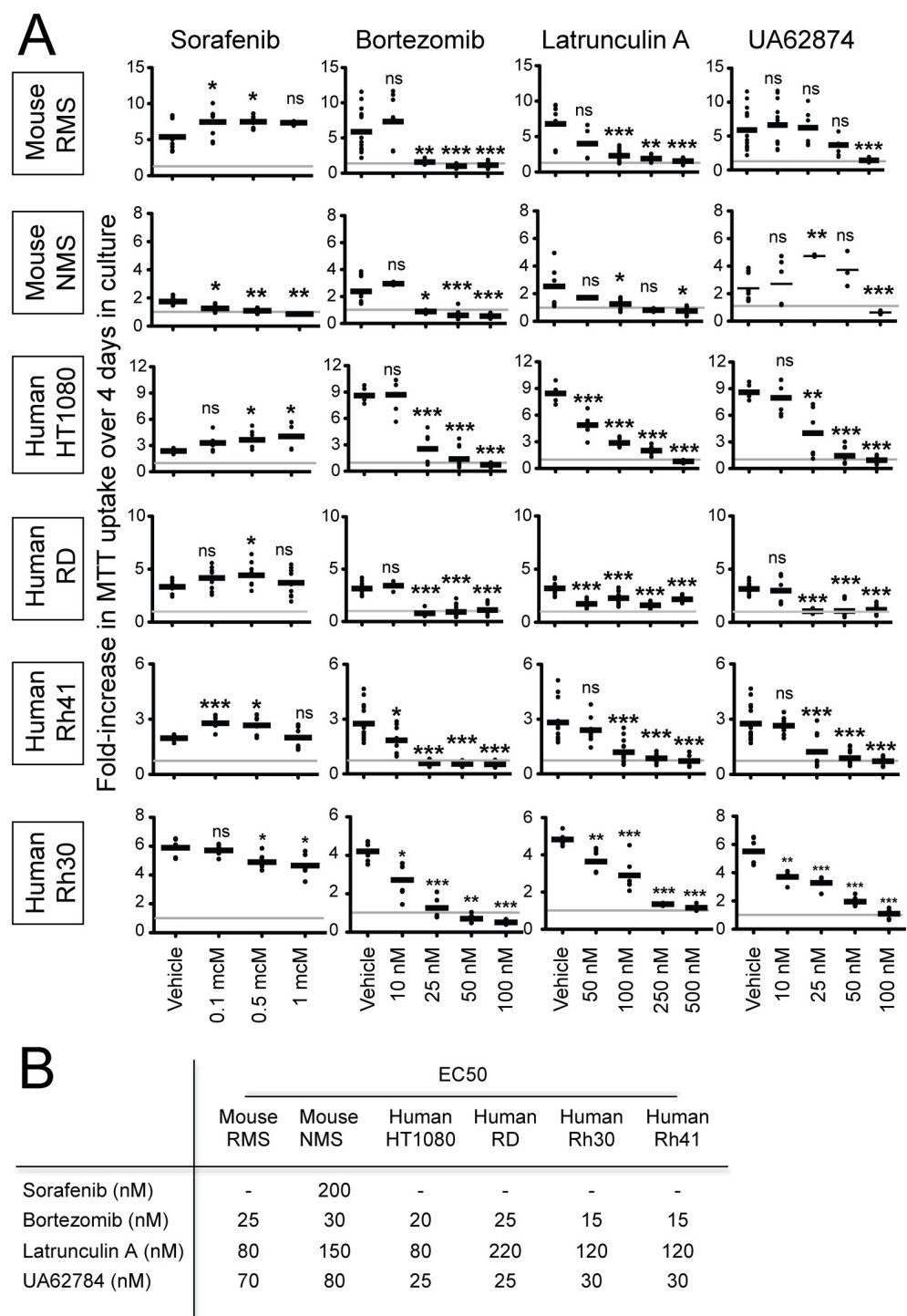

**Fig 3. Chemical inhibition of mouse and human sarcoma cell growth *in vitro*.** (A) Bortezomib, Latrunculin A, and UA62874, previously reported to modulate cellular functions of UBE2C, HAS2 and CENPE, respectively, reduced proliferation of a *Kras;CDKN2A[null]* mouse sarcoma cell line with myogenic differentiation (mouse RMS), a *Kras; CDKN2A[null]* undifferentiated, non-myogenic mouse sarcoma cell line (mouse NMS), the human fibrosarcoma cell line HT1080, the human embryonal RMS cell line RD (PAX3/7:FOXO1-negative) and the human alveolar RMS cell lines Rh30 and Rh41 (PAX3/7:FOXO1-positive). Sorafenib increased growth of the mouse RMS cell line, HT1080, RD and Rh41 (mean +/- SD of 6 technical replicates obtained in 3 independent experiments are presented; ns p≥0.05, * p<0.05, ** p<0.01, *** p <0.001, as determined by T-tests compared to vehicle-treated cells). (B) Estimated EC50 concentrations ranged between 15–30 nM for Bortezomib, 80–150 nM for Latrunculin A and 25–80 nM for UA62784 (EC50 concentrations were calculated using graphpad). Please see also S3 Fig.

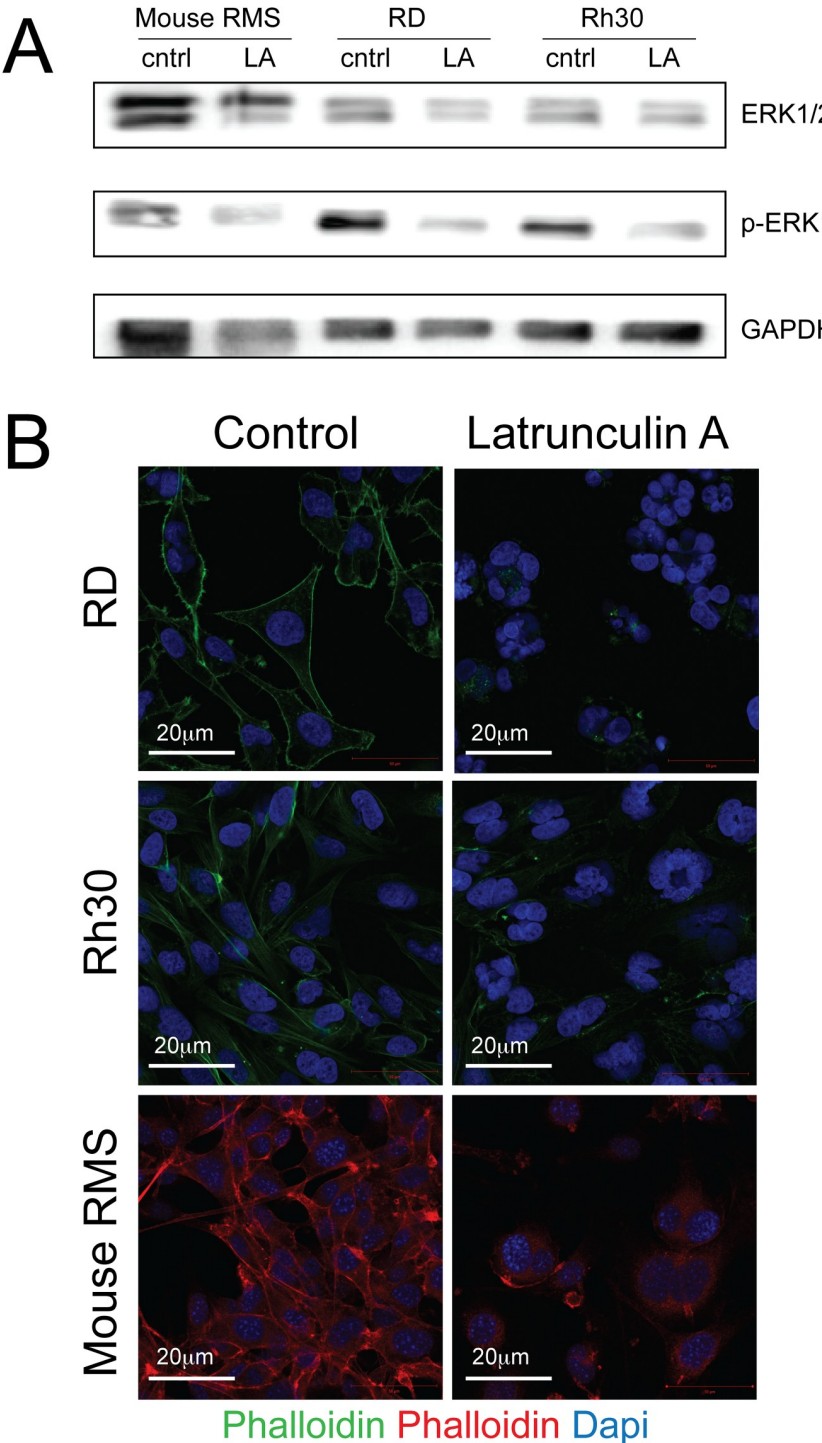

**Fig 4. Protein expression levels of ERK and pERK in human and mouse rhabdomyosarcoma cell lines after treatment with Latrunculin A.** Human RD sarcoma cells were exposed to 250nM latrunculin A, human Rh30 RMS cells to 100nM latrunculin A and mouse RMS sarcoma cells to 100nM latrunculin A. Different concentrations were chosen due to differences in latrunculin A sensitivity between sarcoma cell lines. (A) Evaluation of ERK1/2 (42/44kDa), p-ERK1/2 (22/ 44 kDa) and GAPDH (37 kDa) by Western blotting demonstrated reduced ERK1/2 phosphorylation in latrunculin A-treated sarcoma cells. (B) The F-actin cytoskeleton was evaluated using phalloidin staining. Actin organization was profoundly disrupted in latrunculin A-treated (top right panel)) vs. control (top left panel) RD cells, latrunculin A-treated (middle right panel) vs. control (middle left panel) Rh30 cells and latrunculin A-treated (bottom right panel) vs. control (bottom left panel) mouse RMS cells. Please also see S1 Raw images.

ERK1/2 phosphorylation in latrunculin A-treated sarcoma cells. It was previously shown that disruption of F-actin interrupts growth-factor mediated RAS activation [26]. We speculate that latrunculin A may interfere with RAS pathway activation in RMS by disrupting the actin cytoskeleton. Further studies are needed to fully understand the anti-sarcoma effects of disrupting the actin cytoskeleton by treatment with latrunculin A, its effects on RAS signalling and the anti-RMS efficacy of latrunculin A *in vivo*.

Taken together, this study supports the anti-sarcoma efficacy of latrunculin A and indicates that the multi-tyrosine kinase inhibitor sorafenib should be viewed with caution in the treatment of RMS.

## Supporting information

**S1 Raw images. Original gel images (ERK1/2 western blots).** Original uncropped and unadjusted images of the Western blots examining ERK1/2, p-ERK1/2 and GAPDH in RMS cells are provided.
(PDF)

**S1 Table. Sarcoma genes.** 141 sarcoma-relevant genes were identified by transcriptional profiling of KRAS-driven mouse sarcomas, and their contributions to sarcoma growth were probed by customized shRNA screening. Five candidate genes (marked in bold font) were found to be proliferation-relevant and immediately actionable [5, 6].
(DOCX)

**S1 Fig. Validation of immunohistochemical staining of candidate sarcoma targets.** Staining was established using the following control tissues: Positive control tissues were human colon (UBE2C), human testes (CENPE), human skin (HAS2) and human liver (CREB3L2). Negative control tissues were human brain (UBE2C, CENPE, HAS2) and human colon (CREB3L2).
(TIF)

**S2 Fig. Expression of candidate sarcoma targets in human sarcoma cell lines.** Expression of *Cenpe*, *Has2*, *Creb3l2* and *Ube2c* in human sarcoma cell lines, as well as in human adult and human fetal muscle, was determined by qRT-PCR (Mean +/- SD of 4 technical replicates are presented; ns $p \geq 0.05$, * $p < 0.05$, ** $p < 0.01$, *** $p < 0.001$, as determined by T-tests compared to adult muscle).
(TIF)

**S3 Fig. Expression of candidate sarcoma targets after latrunculin A treatment in human and murine RMS cell lines.** Cells were incubated with Latrunculin A for 96 hours. Expression of *Has2* was determined by qRT-PCR. Latrunculin A treatment did not reduce HAS2 expression (mean +/- SD of 3 technical replicates are presented; ns $p \geq 0.05$, * $p < 0.05$, ** $p < 0.01$, *** $p < 0.001$, as determined by T-tests compared to carrier controls).
(TIF)

## Acknowledgments

The authors thank Alexandra Fischer for administrative support. The authors declare no competing financial interests.

## Author Contributions

**Conceptualization:** Julia Würtemberger, Simone Hettmer.

**Data curation:** Carla Regina.

**Formal analysis:** Carla Regina, Christoph Bauer, Michaela Schneider.

**Investigation:** Julia Würtemberger, Daria Tchessalova, Amy J. Wagers, Simone Hettmer.

**Methodology:** Julia Würtemberger, Daria Tchessalova, Simone Hettmer.

**Project administration:** Simone Hettmer.

**Resources:** Carla Regina, Simone Hettmer.

**Supervision:** Daria Tchessalova, Simone Hettmer.

**Validation:** Michaela Schneider, Simone Hettmer.

**Visualization:** Amy J. Wagers.

**Writing – original draft:** Julia Würtemberger.

**Writing – review & editing:** Simone Hettmer.

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
