## [Decision Letter · Decision Letter 0]

18 Dec 2019

PONE-D-19-29958

Target screening in soft tissue sarcomas.

PLOS ONE

Dear Dr Hettmer,

Thank you for submitting your manuscript to PLOS ONE. After careful consideration, we feel that it has merit but does not fully meet PLOS ONE’s publication criteria as it currently stands. Therefore, we invite you to submit a revised version of the manuscript that addresses the points raised during the review process.

We would appreciate receiving your revised manuscript by Feb 01 2020 11:59PM. To enhance the reproducibility of your results, we recommend that if applicable you deposit your laboratory protocols in protocols.io, where a protocol can be assigned its own identifier (DOI) such that it can be cited independently in the future. For instructions see: http://journals.plos.org/plosone/s/submission-guidelines#loc-laboratory-protocols

We look forward to receiving your revised manuscript.

Kind regards,

Sandro Pasquali, M.D., Ph.D.

Academic Editor

PLOS ONE

Additional Editor Comments (if provided):

Comments from the two reviewers need to be carefully addressed, particularly those from Reviewer #2. Title, introduction, discussion and conclusion should be modified to better reflect the study aim, design and results. The manuscript in the present form is quite misleading, although it can make a contribution to existing literature if properly modified. Please, follow reviewers suggestions.

Journal Requirements:

2. We noticed you have some minor occurrence(s) of overlapping text with the following previous publication(s), which needs to be addressed:

https://doi.org/10.7554/eLife.09436

In your revision ensure you cite all your sources (including your own works), and quote or rephrase any duplicated text outside the Methods section. Further consideration is dependent on these concerns being addressed.

4.  We noted in your submission details that a portion of your manuscript may have been presented or published elsewhere. Please clarify whether this [conference proceeding or publication] was peer-reviewed and formally published. If this work was previously peer-reviewed and published, in the cover letter please provide the reason that this work does not constitute dual publication and should be included in the current manuscript.

5.  Thank you for stating in your Funding Statement:  "This work was funded in part by a Stand Up To Cancer-American Association for Cancer Research Innovative Research Grant (SU2C-AACR-IRG1111; to AJW) and P.A.L.S. Bermuda/St. Baldrick’s (to SH)".

7. Please include your tables as part of your main manuscript and remove the individual files. Please note that supplementary tables (should remain/ be uploaded) as separate "supporting information" files

Reviewers' comments:

Reviewer's Responses to Questions

**Comments to the Author**

1. Is the manuscript technically sound, and do the data support the conclusions?

Reviewer #1: Yes

Reviewer #2: Partly

2. Has the statistical analysis been performed appropriately and rigorously? 

Reviewer #1: Yes

Reviewer #2: N/A

3. Have the authors made all data underlying the findings in their manuscript fully available?

Reviewer #1: Yes

Reviewer #2: No

4. Is the manuscript presented in an intelligible fashion and written in standard English?

Reviewer #1: Yes

Reviewer #2: Yes

5. Review Comments to the Author

Reviewer #1: This is a well designed and well written paper about the possible use of novel treatments for sarcomas.

My only concern is about the results on human tissues. Indeed, the figure 2A is of very poor quality and is difficult to see immunostains. The quthors should provide better photomicrographs to assess the validity of the reactions and the type of expression. Moreover, an H-score or a semi-quantitative evaluation of the expression of the four marker at least in leiomyosarcoma and rhabdomyosarcoma samples should be added.

Reviewer #2: General

This paper focuses on identification of actionable alterations in (rhabdomyo)sarcoma

The authors performed transcriptional profiling on RMS mouse model, displaying myogenic and non-myogenic features and identified 141 candidates; they than screened these candidates with a sh Array they developed and limited the results to the 16 better performing candidates; they than reviewed the literature - unsystematically - for actionable targets, identifying 4/16: UBE2C, CENPE, HAS2 and CREB3L2.

The experimenters then evaluated the expression of these genes by PCR in 9 human cell lines: UBE2C, CENPE and CREB3L2 was significantly overexpressed, in all cell lines, whereas HAS2 was significantly overexpressed in fibrosarcoma cell line, non-rearranged RMS cell line, and in 3/7 rearranged RMS cell lines. They then used FFPE tissue of a commercially available TMA comprising 24 rhabdomyosarcoma (and 26 leiomyosarcoma), to confirm these findings. Although IHC did showed an increase expression between skeletal muscle and RMS cores the magnitude of the effect was not comparable with the qRT-PCT cell culture assays.

They then tested the 4 compounds that should supposedly act on UBE2C, CENPE, HAS2 and CREB3L2 on 4/9 human cell lines and on 2/2 of the mouse derived. In particular, they included Rh30 an outlier in the HAS2 expression. All the drugs, but sorafenib, showed an effect, increasing with the dose. Sorafenib showed a significant effect in human Rh30, and in mouse NMS (2/6). They then focused on the drug (LatrunculinA) that did not showed the target (HAS2) neither in the culture nor in the tissue. The significant expected reduction of HAS2 occurred just in the high-expresser Rh30; they dismissed the results and focused on other possible mechanisms of action of LatrunculinA and introduced (and found) actin organization reduction and impairment of MAPK.

A clear experimental design or a chain of results-hypothesis-testing are lacking and the steps between the sections have some gaps.

There are 3 critical aspects

1) The paper claims to focus on soft tissue sarcoma in general but the majority of experiments are conducted on RMS, which also can often arise outside the so-called soft tissue (i.e. viscera). The story could benefit by focusing on RMS.

2) Experiments are conducted on different cell cultures which display several differences, but some experiments (Anti-proliferative effects of chemical targeting of candidate genes and Latrunculin A effects on sarcoma cells) are performed on different subsets of lines making very difficult to draw conclusions

3) This article finds 4 druggable-genes but the real effect of each drug (on targets or their downstream) is not shown (with the exception of latrunculin A and HAS2), leaving many questions also on the validity of the premises

The paper need major revisions.

Comments on specific sections

TITLE: rather general and non-descriptive of the article content

INTRODUCTION:

Arid.

“RAS pathway genes are frequently mutated in STS”: this statement is not supported by the cited paper, which focuses on RMS alone.

AIMS

Aims are not explicitly stated

METHODS

Methods could benefit some revisions, moreover I failed to find a Supplemental experimental procedures section in the manuscript.

Customized shRNA proliferation screen: Statistical analysis should be incorporated in the main text or the supplementary if relevant for article interpretation.

Immunohistochemistry: time and temperature for the antigen retrieval as well as the ab incubation should also be declared. How the IHC stains on tissue were evaluated?

“Immunocytochemistry”  Immunofluorescence?

RESULTS

“Identification of target genes. A candidate set of 141 sarcoma-relevant genes was identified by transcriptional profiling of genetically engineered mouse sarcomas driven by KRAS(G12v) and CDKN2A deletion (Table S1) (5).” Is not clear if the set of 141 sarcoma relevant genes: is an original finding, is derived from unpublished data of the reference 5 or was already published in the reference 5. Please clarify.

“Sixteen of 141 candidate genes met stringent significance”: “stringent” is a qualifier more suitable for the discussion, then for results.

- Candidate gene expression in sarcomas.

Since the passage between qRT-PCR and patient derived tissue did not showed the same magnitude a WB depicting protein level in the different cell lines could give a link between the two.

- Latrunculin A effects on sarcoma cells

The four targets (UBE2C, CENPE, HAS2 and CREB3L2) were chosen because they should be targets of the four compounds (bortezomib, UA62784, latrunculin A, and sorafenib, respectively) but authors focused on Latrunculin A and its effect on HAS2 expression (already weak in the earlier reported results) and no experiment are displayed to support the effects of the other compound that did showed activity.

“Latrunculin A effects on sarcoma cells. Latrunculin A inhibited the growth of mosue and human sarcoma cell lines”: Typos

Mouse SMP01 cells? Where do they came from? Supposing they are the Mouse RMS they are receiving a dose 25% of their EC50, whereas the RD are a 13% higher and rh30 a 16% lower; however also the GAPDH seems lower in SMP01 LA compared to the control lane.

What is the logical basis to further explore only Latrunculin A and not the other drugs? Why investigate other effects of lantruculin such as ERK phosphorylation?

Discussion

“STS cure depends on radical resection and/or radiation of the tumor.” Reference needed. Moreover, this statement is not supported by the paper cited in the following sentence (14)

“Functional genomic screening of KRAS-driven mouse sarcomas was employed to identify actionable, proliferation-relevant genes of potential therapeutic applicability as anti-sarcoma drugs” the “actionability” of these genes have not been explored.

Relationship with other findings is generic and shortly supported

Potential limitations of the current study are not highlighted

In conclusions the authors suggest to embrace the anti-sarcoma effect of bortezomib, CENPE inhibitors and latrunculin A, whereas the treatment of sarcomas with sorafenib should be viewed with caution because of the negative results on 2/4 human cell lines tested.

6. PLOS authors have the option to publish the peer review history of their article (what does this mean?). If published, this will include your full peer review and any attached files.

Reviewer #1: No

Reviewer #2: Yes: Salvatore Lorenzo Renne

---

## [Author Response · Author response to Decision Letter 0]

18 Mar 2020

Detailed point-by-point response:

The revised manuscript fulfills PLOS ONE’s style requirements. 

2. We noticed you have some minor occurrence(s) of overlapping text with the following previous publication(s), which needs to be addressed:

We have rewritten the relevant sentences and included appropriate citations.

3. PLOS ONE now requires that authors provide the original uncropped and unadjusted images underlying all blot or gel results reported in a submission’s figures or Supporting Information files. 

Original uncropped and unadjusted Western blot images were included as a supplemental figure (see Figure S4, S5).

4. We noted in your submission details that a portion of your manuscript may have been presented or published elsewhere. Please clarify whether this [conference proceeding or publication] was peer-reviewed and formally published. 

This manuscript builds on functional genomic screening of a candidate set of sarcoma genes. The candidate gene set and the screen were published in peer-reviewed journals in 2011 (1) and 2015 (2), respectively. This has been stated in the manuscript, and both articles were referenced. 

Of note, the 2011 paper focuses on the studies that led to identifying the gene set as a whole. The 2015 paper describes the screen and investigates its top hit (ASNS).

This current manuscript investigates 4 candidate genes, which belong to the candidate gene set and were included in the screen. Neither the 2011 nor the 2015 paper include an in-depth investigation or discussion of these candidates.

5. Funding statement.

This work was funded by a Stand Up To Cancer-American Association for Cancer Research Innovative Research Grant (SU2C-AACR-IRG1111; to AJW) and P.A.L.S. Bermuda/St. Baldrick’s (to SH). We did not receive any other external funding to support this study. The funding statement was modified accordingly. 

6. Phrase “data not shown”.

The revised manuscript doesn’t include the phrase “data not shown”. All relevant data are provided within the paper, and there are no references to inaccessible data.

7. Tables.

Table1 was included in the revised, main manuscript. Table S1 was provided as a separate supporting file.

Reviewer #1:

This is a well-designed and well written paper about the possible use of novel treatments for sarcomas.

We thank the reviewer for the positive feedback.

My only concern is about the results on human tissues. Indeed, the figure 2A is of very poor quality and is difficult to see immunostains. The quthors should provide better photomicrographs to assess the validity of the reactions and the type of expression. Moreover, an H-score or a semi-quantitative evaluation of the expression of the four marker at least in leiomyosarcoma and rhabdomyosarcoma samples should be added.

We apologize for the poor quality of figure 2A. We have provided new photomicrographs to better demonstrate the expression of the candidate genes in human sarcoma tissue. Also, we provided further details on how immunohistochemistry staining was evaluated. Specifically, CENPE (nuclear), UBE2C (cytoplasmic), CREB3L2 (cytoplasmic) and HAS2 (nuclear) staining was evaluated by two independent operators. If > 25% of cells per core exhibited a positive signal, antigen expression was considered positive.

Reviewer #2:

This paper focuses on identification of actionable alterations in (rhabdomyo)sarcoma. The authors performed transcriptional profiling on RMS mouse model, displaying myogenic and non-myogenic features and identified 141 candidates; they then screened these candidates with a sh Array they developed and limited the results to the 16 better performing candidates; they than reviewed the literature - unsystematically - for actionable targets, identifying 4/16: UBE2C, CENPE, HAS2 and CREB3L2.

The experimenters then evaluated the expression of these genes by PCR in 9 human cell lines: UBE2C, CENPE and CREB3L2 was significantly overexpressed, in all cell lines, whereas HAS2 was significantly overexpressed in fibrosarcoma cell line, non-rearranged RMS cell line, and in 3/7 rearranged RMS cell lines. They then used FFPE tissue of a commercially available TMA comprising 24 rhabdomyosarcoma (and 26 leiomyosarcoma), to confirm these findings. Although IHC did showed an increase expression between skeletal muscle and RMS cores the magnitude of the effect was not comparable with the qRT-PCT cell culture assays. They then tested the 4 compounds that should supposedly act on UBE2C, CENPE, HAS2 and CREB3L2 on 4/9 human cell lines and on 2/2 of the mouse derived. In particular, they included Rh30 an outlier in the HAS2 expression. All the drugs, but sorafenib, showed an effect, increasing with the dose. Sorafenib showed a significant effect in human Rh30, and in mouse NMS (2/6). They then focused on the drug (LatrunculinA) that did not showed the target (HAS2) neither in the culture nor in the tissue. The significant expected reduction of HAS2 occurred just in the high-expresser Rh30; they dismissed the results and focused on other possible mechanisms of action of LatrunculinA and introduced (and found) actin organization reduction and impairment of MAPK.

A clear experimental design or a chain of results-hypothesis-testing are lacking and the steps between the sections have some gaps.

We thank the reviewer for thoroughly reviewing the manuscript and providing valuable feedback, which improved the manuscript substantially. 

The paper claims to focus on soft tissue sarcoma in general but the majority of experiments are conducted on RMS, which also can often arise outside the so-called soft tissue (i.e. viscera). The story could benefit by focusing on RMS.

The manuscript was rewritten to focus on RMS.

Experiments are conducted on different cell cultures which display several differences, but some experiments (Anti-proliferative effects of chemical targeting of candidate genes and Latrunculin A effects on sarcoma cells) are performed on different subsets of lines making very difficult to draw conclusions.

We chose to use a wide array of different sarcoma cell lines to test the effects of candidate chemicals. Subsequent experiments, aimed at evaluating target gene expression and latrunculin effects, employed mouse RMS cells and the human RD and Rh30 lines. We respectfully point out that the latter three lines were tested as part of all experiments. This is in keeping with the RMS focus suggested by the reviewer. 

This article finds 4 druggable-genes but the real effect of each drug (on targets or their downstream) is not shown (with the exception of latrunculin A and HAS2), leaving many questions also on the validity of the premises.

We chose to focus our studies on the effects of latrunculin A on sarcoma cells. Sorafenib (targeting Creb3l2), bortezomib (targeting Ube2c) and UA62874 (targeting Cenpe) were not included in further validation experiments, because (i) sorafenib did not reduce sarcoma cell proliferation, and (ii) the published in vivo effects of all three drugs on sarcoma xenograft growth were discouraging (3-5). This was explained in the discussion of the revised manuscript.

TITLE: rather general and non-descriptive of the article content

We appreciate the reviewer’s critique. The revised manuscript is entitled “Growth inhibition associated with disruption disruption of the actin cytoskeleton by Latrunculin A in rhabdomyosarcoma cells“ to reflect the focus of the experiments on latrunculin A effects in rhabdomyosarcoma. 

INTRODUCTION: Arid. “RAS pathway genes are frequently mutated in STS”: this statement is not supported by the cited paper, which focuses on RMS alone.

RAS pathway genes were found to be mutated in RMS and other types of STS. An additional reference (6) was included to support RAS mutations in non-RMS STS.

AIMS: Aims are not explicitly stated.

The introduction was rephrased to clearly state the aims of the study.

METHODS: Methods could benefit some revisions, moreover I failed to find a Supplemental experimental procedures section in the manuscript.

To provide readers with easier access to the technical details, all experimental procedures were included within the main manuscript. As suggested by the reviewer, the experimental procedures were further expanded, and details of the statistical and IHC analyses were clarified in the revised manuscript.

Customized shRNA proliferation screen: Statistical analysis should be incorporated in the main text or the supplementary if relevant for article interpretation.

The statistical analysis of the customized shRNA proliferation screen was published in detail in 2015 (2). In the revised manuscript, we included a brief description of the statistics, which established growth-relevant candidate genes.

Immunohistochemistry: time and temperature for the antigen retrieval as well as the ab incubation should also be declared. How the IHC stains on tissue were evaluated? “Immunocytochemistry”  Immunofluorescence?

Antigen expression was evaluated by standard immunohistochemistry procedures. Binding of primary antibodies by target antigens was detected by labeling with biotinylated secondary antibodies and Streptavidin-HRP. Slides were then exposed to DAB substrate, which reacts with HRP to produce a brown-colored signal. The brown-colored signal was evaluated by light microscopy. The IHC section in the revised manuscript was expanded to include further technical details, including time and temperature of the antigen retrieval. 

With respect to the evaluation of IHC staining results, CENPE (nuclear), UBE2C (cytoplasmic), CREB3L2 (cytoplasmic) and HAS2 (nuclear) staining was evaluated by two independent operators. If > 25% of cells per core exhibited a positive signal, antigen expression was considered positive. Please also see our response to the comments made by reviewer #1. 

RESULTS: “Identification of target genes. A candidate set of 141 sarcoma-relevant genes was identified by transcriptional profiling of genetically engineered mouse sarcomas driven by KRAS(G12v) and CDKN2A deletion (Table S1) (5).” Is not clear if the set of 141 sarcoma relevant genes: is an original finding, is derived from unpublished data of the reference 5 or was already published in the reference 5. Please clarify.

We apologize for the ambiguity. The revised manuscript clearly indicates that the set of 141 candidate genes was previously identified and published (1). 

“Sixteen of 141 candidate genes met stringent significance”: “stringent” is a qualifier more suitable for the discussion, then for results.

We agree with the reviewer and eliminated the word “stringent”.

Candidate gene expression in sarcomas: Since the passage between qRT-PCR and patient derived tissue did not showed the same magnitude a WB depicting protein level in the different cell lines could give a link between the two.

We respectfully note that discrepancies between high-passage sarcoma cell lines and primary tissue are common. This was stated in the revised manuscript.

Latrunculin A effects on sarcoma cells: The four targets (UBE2C, CENPE, HAS2 and CREB3L2) were chosen because they should be targets of the four compounds (bortezomib, UA62784, latrunculin A, and sorafenib, respectively) but authors focused on Latrunculin A and its effect on HAS2 expression (already weak in the earlier reported results) and no experiment are displayed to support the effects of the other compound that did showed activity. What is the logical basis to further explore only Latrunculin A and not the other drugs? Why investigate other effects of lantruculin such as ERK phosphorylation?

As discussed above, we chose to focus our studies on the effects of latrunculin A on sarcoma cells. Sorafenib (targeting Creb3l2), bortezomib (targeting Ube2c) and UA62874 (targeting Cenpe) were not included in further validation experiments, because (i) sorafenib did not reduce sarcoma cell proliferation, and (ii) the published in vivo effects of all three drugs on sarcoma xenograft growth were discouraging (3-5). This was explained in the discussion of the revised manuscript.

“Latrunculin A effects on sarcoma cells. Latrunculin A inhibited the growth of mosue and human sarcoma cell lines”: Typos

We apologize for the typo. It was corrected.

Mouse SMP01 cells? Where do they came from? Supposing they are the Mouse RMS they are receiving a dose 25% of their EC50, whereas the RD are a 13% higher and rh30 a 16% lower; however also the GAPDH seems lower in SMP01 LA compared to the control lane.

In everyday lab practice, we use the name SMP01 to refer to the strain of mouse Kras;CDKN2Anull mouse RMS cells (1), which was used for the experiments in this study. The term SMP01 was removed from the revised manuscript and replaced with mouse RMS cells. 

DISCUSSION: “STS cure depends on radical resection and/or radiation of the tumor.” Reference needed. Moreover, this statement is not supported by the paper cited in the following sentence (14).

The paragraph on sarcoma treatment was rewritten. We added another reference (7) to support that successful treatment of soft-tissue sarcomas depends on adequate local control.

“Functional genomic screening of KRAS-driven mouse sarcomas was employed to identify actionable, proliferation-relevant genes of potential therapeutic applicability as anti-sarcoma drugs” the “actionability” of these genes have not been explored. 

We agree with the reviewer. The word “actionable” was removed. Further studies are needed to examine the therapeutic applicability of latrunculin A in RMS. The latter was stated in the revised manuscript.

Relationship with other findings is generic and shortly supported Potential limitations of the current study are not highlighted.

We thank the reviewer for this important comment. Limitations of the study were discussed in the revised manuscript. 

References

1. Hettmer S, Liu J, Miller CM, Lindsay MC, Sparks CA, Guertin DA, et al. Sarcomas induced in discrete subsets of prospectively isolated skeletal muscle cells. Proc Natl Acad Sci U S A. 2011;108(50):20002-7.

2. Hettmer S, Schinzel AC, Tchessalova D, Schneider M, Parker CL, Bronson RT, et al. Functional genomic screening reveals asparagine dependence as a metabolic vulnerability in sarcoma. Elife. 2015;4.

3. Houghton PJ, Morton CL, Kolb EA, Lock R, Carol H, Reynolds CP, et al. Initial testing (stage 1) of the proteasome inhibitor bortezomib by the pediatric preclinical testing program. Pediatr Blood Cancer. 2008;50(1):37-45.

4. Keir ST, Maris JM, Lock R, Kolb EA, Gorlick R, Carol H, et al. Initial testing (stage 1) of the multi-targeted kinase inhibitor sorafenib by the pediatric preclinical testing program. Pediatr Blood Cancer. 2010;55(6):1126-33.

5. Lock RB, Carol H, Morton CL, Keir ST, Reynolds CP, Kang MH, et al. Initial testing of the CENP-E inhibitor GSK923295A by the pediatric preclinical testing program. Pediatr Blood Cancer. 2012;58(6):916-23.

6. Yoo J, Robinson RA, Lee JY. H-ras and K-ras gene mutations in primary human soft tissue sarcoma: concomitant mutations of the ras genes. Mod Pathol. 1999;12(8):775-80.

7. Crago AM, Brennan MF. Principles in Management of Soft Tissue Sarcoma. Adv Surg. 2015;49:107-22.

---

## [Decision Letter · Decision Letter 1]

26 Jun 2020

PONE-D-19-29958R1

Growth inhibition associated with disruption of the actin cytoskeleton by Latrunculin A in rhabdomyosarcoma cells.

PLOS ONE

Dear Dr. Hettmer,

Thank you for submitting your manuscript to PLOS ONE. After careful consideration, we feel that it has merit but does not fully meet PLOS ONE’s publication criteria as it currently stands. Therefore, we invite you to submit a revised version of the manuscript that addresses the points raised during the review process.

We look forward to receiving your revised manuscript.

Kind regards,

Sandro Pasquali, M.D., Ph.D.

Academic Editor

PLOS ONE

Additional Editor Comments (if provided):

Comemnts have been adressed.

Please review the paper again for typos. The acceptance of the paper depends on these corrections. If the corrections will be made and the supplementary file provided, the manuscript will be accepted. Apologies for keep you waiting, the COVID pandemic has made our working schedule very busy.

Reviewers' comments:

Reviewer's Responses to Questions

**Comments to the Author**

1. If the authors have adequately addressed your comments raised in a previous round of review and you feel that this manuscript is now acceptable for publication, you may indicate that here to bypass the “Comments to the Author” section, enter your conflict of interest statement in the “Confidential to Editor” section, and submit your "Accept" recommendation.

Reviewer #1: All comments have been addressed

Reviewer #2: All comments have been addressed

2. Is the manuscript technically sound, and do the data support the conclusions?

Reviewer #1: Yes

Reviewer #2: Yes

3. Has the statistical analysis been performed appropriately and rigorously? 

Reviewer #1: Yes

Reviewer #2: Yes

4. Have the authors made all data underlying the findings in their manuscript fully available?

Reviewer #1: Yes

Reviewer #2: Yes

5. Is the manuscript presented in an intelligible fashion and written in standard English?

Reviewer #1: Yes

Reviewer #2: Yes

6. Review Comments to the Author

Reviewer #1: All the comments have been addressed by the authors. The paper is now worth of publication without further modifications.

Reviewer #2: There are few typos, for example in the methods section, paragraph "Customized shRNA proliferation screen": " T-tests and the algorithm published by the algorithm published by J. W.McNicol and G. Hogan (11)."

Moreover I cannot find the " Supplemental Experimental Procedures ", please contact the editor.

7. PLOS authors have the option to publish the peer review history of their article (what does this mean?). If published, this will include your full peer review and any attached files.

Reviewer #1: No

Reviewer #2: **Yes: **Salvatore Lorenzo Renne

---

## [Author Response · Author response to Decision Letter 1]

12 Jul 2020

Dear Dr. Pasquali,

I am pleased to re-submit the attached, revised manuscript for consideration for publication in PLOS ONE.

As you know, the manuscript investigates expression of four proliferation-relevant genes in sarcoma cell lines and tissue and examines their potential therapeutic applicability. We demonstrate that disruption of the actin cytoskeleton in sarcoma cells by latrunculin A is associated with a reduction in RMS cell growth.

We are grateful to the reviewers and to you for the constructive critiques, which improved the manuscript substantially. Changes in the manuscript have been marked up, and a detailed point-by-point response has been included below. 

I look forward to hearing from you soon. 

Sincerely,

PD Dr. Simone Hettmer

Detailed point-by-point response:

Changes were marked uisng the track changes function in the revised manuscript.

Reviewer #1: All the comments have been addressed by the authors. The paper is now worth of publication without further modifications.

We thank the reviewer for the positive feedback.

Reviewer #2: There are few typos, for example in the methods section, paragraph "Customized shRNA proliferation screen": " T-tests and the algorithm published by the algorithm published by J. W.McNicol and G. Hogan (11)."

We apologize for the typos. They were corrected.

Moreover I cannot find the " Supplemental Experimental Procedures ", please contact the editor.

To provide readers with easier access to the technical details, all experimental procedures were included within the main manuscript. The reference to Supplemental Experimental Procedures was removed. We apologize for placing this misleading sentence in the previous versions of the manuscript.

---

## [Editor Report · Decision Letter 2]

20 Aug 2020

Growth inhibition associated with disruption of the actin cytoskeleton by Latrunculin A in rhabdomyosarcoma cells.

PONE-D-19-29958R2

Dear Dr. Hettmer,

We’re pleased to inform you that your manuscript has been judged scientifically suitable for publication and will be formally accepted for publication once it meets all outstanding technical requirements.

Kind regards,

Sandro Pasquali, M.D., Ph.D.

Academic Editor

PLOS ONE

---

## [Editor Report · Acceptance letter]

25 Aug 2020

PONE-D-19-29958R2 

Growth inhibition associated with disruption of the actin cytoskeleton by Latrunculin A in rhabdomyosarcoma cells. 

Dear Dr. Hettmer:

I'm pleased to inform you that your manuscript has been deemed suitable for publication in PLOS ONE. Congratulations! Your manuscript is now with our production department. 

Kind regards, 

on behalf of

Dr. Sandro Pasquali 

Academic Editor

PLOS ONE